

# Ultra-low-cost manual soil respiration chamber

Bartosz M. Zawilski and Vincent Bustillo

CESBIO Université de Toulouse, CNES, CNRS, INRA, IRD, UT3-Paul Sabatier, Toulouse, 31000 France

*Correspondence to: Bartosz M. Zawilski (bartosz.zawilski@cnrs.fr)*

**Abstract.** Soil respiration measurement is important to assess natural carbon dioxide production. The closed chamber technique allows relatively easy soil respiration monitoring. A planned spatially large-scale campaign incites us to implement our ultra-low-cost portative chamber The chamber itself is entirely built from commercial parts with little, easy-to-make, quick

machining work. The resulting setup is an easy-to-operate, standalone, robust device. The used sensors are cost-effective yet accurate digital sensors that were successfully checked against some reference sensors. All these characteristics made the described chamber accessible to build and use for a wide scientific and educational community. The wide interest aroused by this construction among our colleagues pushes us to share our achievements. In this short note, we describe this simple device along with its sensors and apparent respiration quotient tip.

## 1 Background

On average, the soil has nearly double the amount of carbon dioxide ($CO_2$) than in the terrestrial atmosphere (Smith 2012). Furthermore, it is one of the biggest generators of $CO_2$, and the frost-free soil generates nearly ten times as much $CO_2$ as the whole amount of fossil fuels burned by humanity. Due to the increased microbial activity brought on by the increased soil temperature, this natural $CO_2$ production is increasing by around 0.1 percent per year (Bond-Lamberty and Thomson 2010).

In the context of global warming due to the increase in atmospheric greenhouse gas concentrations, such as $CO_2$, particular attention is given to soil respiration. There are several techniques and sub-techniques to achieve this goal. One of the most widespread techniques is the closed-chamber technique. This technique is about a century old (Bornemann, 1920), but it has been continuously improved and allows us to monitor more GHG. Among closed chambers, we can distinguish automatic

chambers and manual chambers. Each technique has its pros and cons (Savage et al., 2003; Yao et al., 2008; Lee, 2018). The automatic chambers allow it to operate automatically, which is a salvatory relief, allowing a relatively high operation rate, even during the night. However, the cost and complexity of these chambers prevent their large spread, leading to a relatively high incertitude when spatial variability is important.

Manually operated chambers rely on the same principle except that the chamber operations (closure and opening) are manual

and require a human presence. This kind of chamber can hardly be used during the night, in rain, or in any meteorological



condition that could make human presence exhausting. However, a punctual measurement can be done on a large spatial scale without any external power supply. Not only are these chambers portable, allowing the use of only one chamber in several locations, but the cost of the manually operated chambers is much less important compared to the automatic chambers. The lower this cost, the more important its duplication possibility, allowing a quick and large measurement campaign by a scientific

group or for educational purposes.

This short note describes an ultra-low-cost (200 USD for the basic configuration) and fast construction (1-2 weeks) chamber using ultra-low-cost, yet accurate, sensors (20 to 100 USD for the basic sensors excluding oxygen sensors). The described chamber is built using only some commercially available parts requiring only a little machining work, which is so-called MacGyver science. (Hut et al., 2020). Also, to complete our knowledge about soil respiration, oxygen sensors were

implemented, and their functioning will be described.

## 2 Materials and Method

For the needs of large-scale spatial soil respiration measurements, a portable chamber with detachable stainless-steel collars to be placed in the soil was built. Cost-effective construction was sought but did not impair the quality of the measurements. As mentioned by numerous authors, the chamber needs to have its internal air mixed by a fan; proper sealing between the

chamber and the soil is essential, along with a pressure equilibration device (Koskinen and Mosie, 1981; Parkin and Venterea, 2010; Christiansen et al., 2011; Clough et al., 2013).

### 2.1 Electronic modules composing the data logger

The manual soil respiration chamber described here employs a datalogger made of commercial electronics modules for querying and logging information gathered by several sensors. The entire set of modules is housed in a handheld enclosure

along with a GPS antenna. A basic UART TTL bus-attached GPS antenna was also incorporated because the same chamber is utilized in multiple locations to help track the precise location of the measurement (Fig. 1).

To read and log data from sensors, a data logger was built with a generic clone of an Arduino Mega Pro for its multiple digital buses (I²C, SPI, and UART), its multiple hardware UART serial ports, and its compactness. The real-time is provided by a generic Real Time Clock module (RTC) powered by a precise DS3231 chip on an I2C bus from Dallas Semiconductor (Dallas,

Texas, USA), owned nowadays by Analog Devices (Wilmington, Massachusetts, USA). A generic μSD card reader on the SPI bus ensures data-saving ability. In the case of RS-232 bus use, a generic module based on an MAX3232 chip from Maxim Integrated (San Jose, California, USA), also owned by Analog Devices, converts the RS-232 level to the TTL level. A small fan (MC20100V3-Q01U-G99, 5V, 0.33W, 20x20x10mm, MagLev from Sunon Electric Machine Industry Company Limited, Qianzhen, District Kaohsiung, Taiwan) fixed on a light and holed stainless-steel plate inside the cloche is gently mixing internal



air during the measurement cycle. MagLev (magnetic levitation) life span is very long: 100000 h; the rotating speed is relatively

slow (11000 RPM); the rated air flow is 1.2 CFM; and the static pressure is less than 45 Pa.

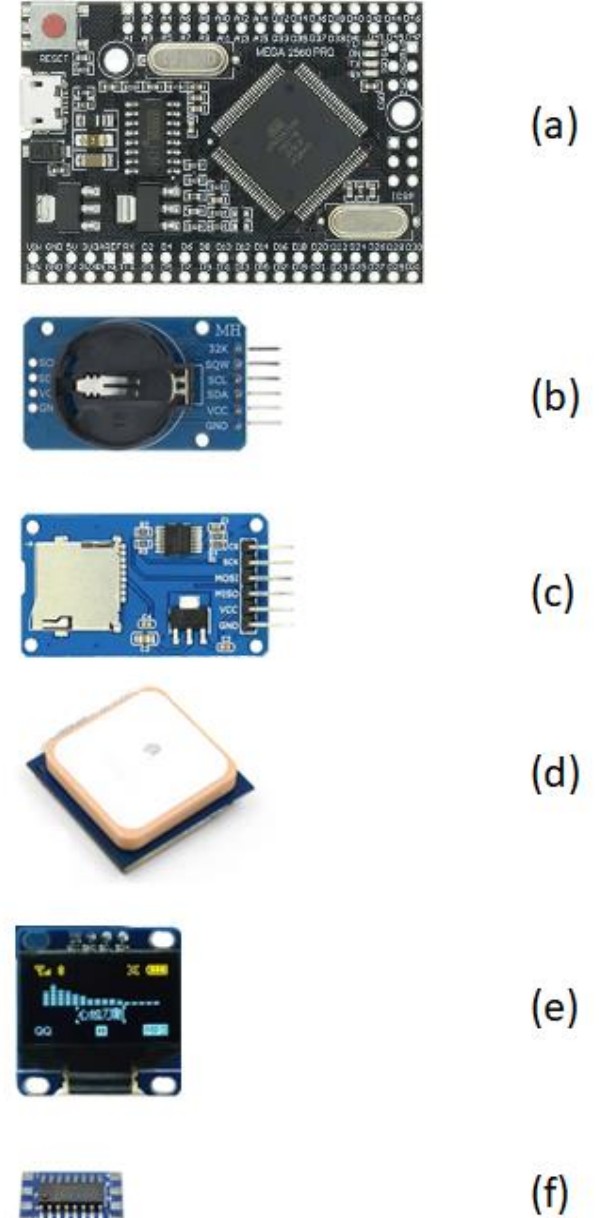

**Figure 1. Data logger built with a) Arduino Mega Pro, b) RTC, c) µSD card reader, d) GPS, e) OLED display, f) RS232 to TTL**
**module.**



A generic 0.96", two-color I²C OLED display allows to indicate all useful information, such as the µSD card state, GPS position reading, logger state, or battery charge level, along with current sensor readings and acquisition time (Fig. 2).

All electronics modules are housed inside a handheld enclosure (Fig. 3) that also contains a generic power bank module using two 18650 lithium-ion rechargeable batteries. The power bank filled with two generic batteries allows 12 hours of uninterrupted operation. A USB socket and short cable allow charging the batteries using a generic USB charger but also to establish a link with the Arduino to program it or to withdraw the µSD stored data without having to dismount the µSD card. Three waterproof push buttons allow the operation of all electronics modules (power On/Off, GPS coordinate memorization,

internal fan operation, and launch/stop a measuring cycle). Two waterproof cable glands allow for safe passage of the sensors and fan cables.

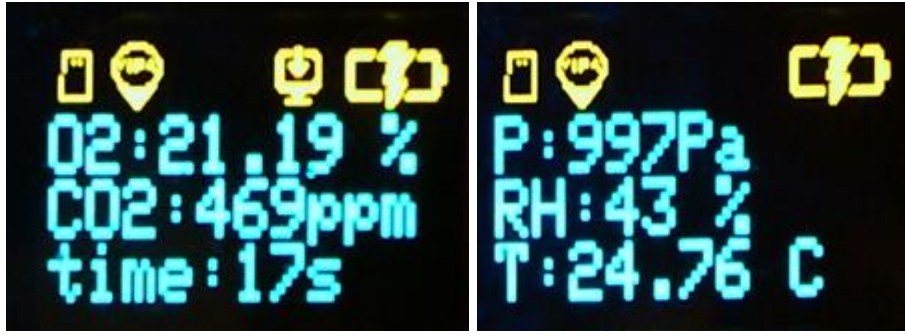

**Figure 2. small 0.96", yet very readable, OLED display permuting screens with different useful information.**

**2.2 Body of the chamber**

The body of the device (Fig. 4) is built around a sanitary stainless-steel Triclover (also called Triclamp) dome reducer (6"–2"). The 2" opening is end-cap obturated and clamped with a 2" Triclover bracket and its Polytetrafluoroethylene (PTFE) joint. The 2" end-cap is pierced for one waterproof cable gland with 5V power cable, I²C bus, and, eventually, UART bus cable. A second hole may be drilled for another cable gland that may be needed for an optional OXYBase oxygen sensor, and the third

hole is for a small exhaust porous silencer used for the equilibration of internal air pressure with external air pressure during the measurement cycle. Two other small holes, tapered for M3 screws, are destined to hold two spacers on which the internal plateau is screwed. The second 6" opening is clamped during the operation on a stainless-steel collar previously placed into the soil at the chosen location. Again, a Triclover bracket and its PTFE joint, but 6" this time, ensure a correct sealing between the chamber body and the collar. The collar itself is made from a 6" Triclover lathe-sharpened ferrule.



(d)

(a)

(b)

(e)

(c)


**Figure 3. The enclosure is made from a) a handheld enclosure, c) µ-USB socket with a short cable, b) a power bank, d) two waterproof cable glands, e) three waterproof push buttons (one black self-locking On-Off and two momentary (On)-Off: one red, one black).**



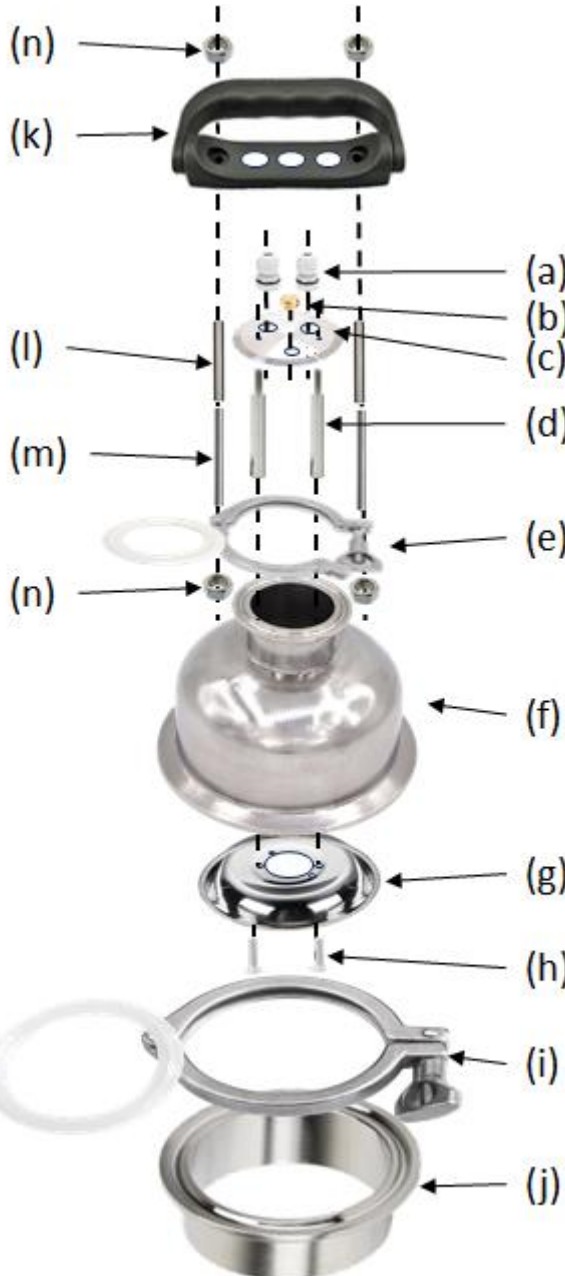

**Figure 4. The body of the chamber is built with a) two waterproof cable glands, b) a small porous pneumatic silencer, c) 2" stainless-steel end-cap, d) two M3 spacers, e) 2" Triclover bracket with a PTFE joint, f) 6" to 2" Triclover reducer, g) small stainless-steel plate, h) two M3 screws, i) 6" Triclover bracket with PTFE joint, j) 6" stainless-steel bottom-sharpened fitting, k) Pierced plastic handle, l) stainless-steel tube partially covering the: m) M4 threaded stainless-steel rod, n) stainless-steel M4 nuts.**




The 2" Triclover bracket studs were removed and replaced with a piece of stainless-steel M4 threaded rods partially covered

with matching tube. On each end, the rods are bolted, at the bottom to the bracket and on the top to the handle.

The handle was drilled to hold three push buttons on the upper side and the handheld enclosure on the bottom.

Figure 5 shows the overall finished setup.

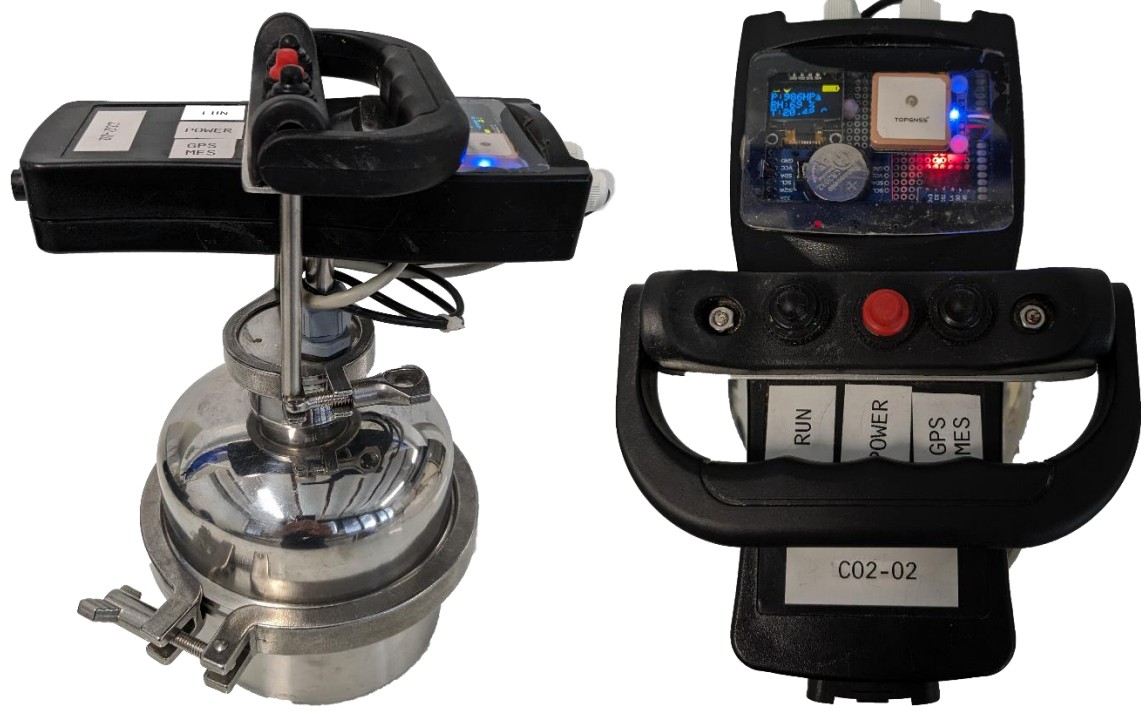


**Figure 5. Side-view of the assembled chamber positioned on the collar and Top-view of the chamber.**

This chamber has an internal cloche volume ($V$ = 1.65 dm³) to internal collar surface ($S$ = 2.01 dm²) ratio of $R$ = 0.82 dm. Of

course, when measurements are computed, the volume of air comprised between the soil surface and the top of the collar has

to be added to the cloche volume.

**2.3 Embedded sensors and fan**

The embedded sensors are the heart of the device. Nowadays, some miniaturized devices allow precise sensing of numerous

physical quantities. The air pressure, temperature, and humidity can be precisely monitored by a minuscule BME280 sensor

(Bosch Sensortec GmbH, Reutlingen, Germany).





Gas concentration monitoring can be achieved with any small and accurate enough sensor. Several techniques are available, such as semiconductor, electrochemical, or optical. We do not embed a methane ($CH_4$) sensor, but this is a possibility using a semiconductor sensor (Riddick et al., 2020; Bastviken et al., 2020; Furst et al., 2021).

For $CO_2$ concentration monitoring, non-dispersive infrared (NDIR) sensors are currently used (Hodgkinson et al., 2013; Dinh
et al., 2016). They are relatively cost-effective, small, and accurate enough. Not only $CO_2$ can be monitored with the NDIR sensors, but also some other gases, such as carbon monoxide (Diharja et al., 2012). Other miniaturized sensors can be used for CO2, but we found the NDIR sensors have the best quality-to-cost ratio for $CO_2$ measurement.

Precise oxygen depletion measurement is challenging as the main atmospheric oxygen concentration (20.9%) is relatively high
compared to the concentration variations in the closed chamber. When the $CO_2$ concentration can be multiplied by 5 after a few minutes inside a closed chamber, the oxygen concentration decreases only by barely a few percent of the initial concentration. Then, the sensor dedicated to the oxygen concentration measurement should be particularly accurate and stable. For this reason, we chose to work with optical sensors such as LuminOx and OXYBase. Both are optical, and their functioning is close to each other. The LuminOx (SST Sensing Ltd., 5 Hagmill Crescent, Shawhead Industrial Estate, Coatbridge, UK) is
based on non-depleting luminescence technology, and the OXYBase (PreSens-Precision Sensing GmbH, Regensburg, Germany) is based on quenching luminescence. The absolute accuracy, resolution, and response time of the OXYbase sensor is better than that of the LuminOx sensor. The OXYbase sensor costs over six times as much as LumiOx as the cost is substantially non-linear with accuracy. We then have to choose based on our goal. The oxygen sensors are, by far, the most expensive sensors on this device (100 USD–650 USD). Oxygen depletion measurement is interesting and brings new insights
into soil respiration (Turcu et al., 2005; Helm et al., 2021); however, their use is still optional.

used, and some of the existing sensor's specifications are summarized in Table 1. Notice that the provided specifications apply to room temperature and pressure ranges. Some of the sensors have several possible configurations, providing different measurement units, and so on. However, in Table 1, for the sake of clarity, only one of the possible configurations is given.






| Sensor model | Brand | Main mesure - unit | Bus | Range | Accuracy | Resolution | Response Time | Remarks |
|---|---|---|---|---|---|---|---|---|
| BME280 | Bosch | P hPa | I²C and SPI | 300 to 1100 hPa | ± 0.12 hPa | 0.18 Pa | Faster than RH | Offset ±1.5 Pa/K |
| | | T °C | | -40 to+85 °C | ± 0.5 °C | 0.01 °C | Faster than RH | |
| | | RH % | | 0 to 100 % | ± 3 % | 0.008 % | $\tau_{63} = 1$ s | 1% hysteresis |
| MH-Z16 | Winsen | $CO_2$ ppm | UART | 0 to 2000 ppm | 50 ppm +5% of reading | 1 ppm | $\tau_{90} < 60$ s | Self-calibrated |
| CozIR | SST | $CO_2$ ppm | I²C and UART | 0 to 2000 ppm | 30 ppm + 3% of reading | 1 ppm | $\tau_{90} < 30$ s | Offset 0.14% of reading per 1 mbar barometric pressure change from 1013 mbar |
| SCD30 | Sensirion | $CO_2$ ppm | I²C and UART | 400 to 10000 ppm | 30 ppm + 3% of reading | 1 ppm | $\tau_{63} < 20$ s | Measures also RH and T. Self-calibrated |
| LuminOx | SST | $O_2$ % | UART | 0 to 25% of $O_2$ | 2% FS (0.5% of $O_2$) | 0.01% of $O_2$ | $\tau_{90} < 30$ s | Measures also P |
| OXYbase | PreSens | $O_2$ hPa | RS-232 RS-485 4-20mA | 0 to 500 hPa | 4 hPa at 200 hPa | 0.3 hPa at 200 hPa | $\tau_{90} < 10$ s | Measures also dissolved $O_2$ |

**Table 1. The mentioned measured parameters are: Pressure (P), temperature (T), relative air humidity (RH), carbon dioxide concentration ($CO_2$), oxygen concentration ($O_2$)**



Figure 6 shows the used sensors and Fan embedded under the cloche.

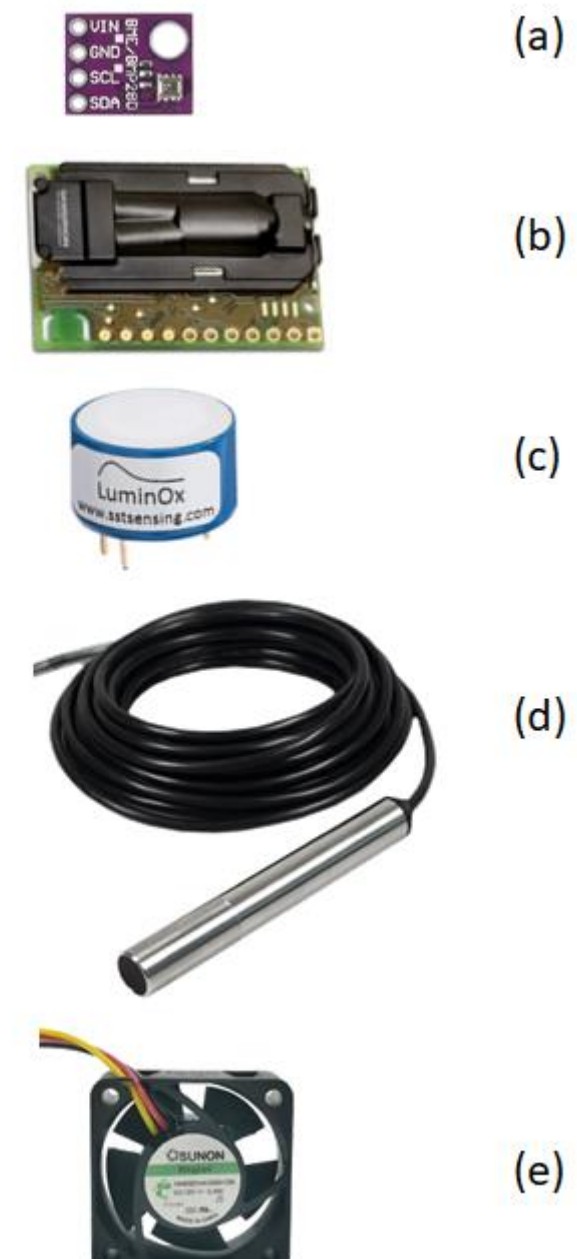

**Figure 6. Used sensors and fan: a) BME280, (P, T, and RH), b) SCD30 ($CO_2$), LuminOx ($O_2$), d) OXYBase ($O_2$), e) Maglev fan.**

All gas-analyzing sensors are digital and placed under a cloche on a dedicated prototype PCB held by the fan. The embedded fan is gently mixing the air entrapped under a closed cloche to homogenize it as well as possible without provoking pressure





pulsations that may affect measured effluxes (Le Dantec et al., 1999; Koskinen et al., 2014). The semi-spherical shape of the cloche helps to prevent poorly mixed areas (Livingston and Hutchinson, 1995).

BME280 is used to measure air pressure, temperature, and humidity. Air humidity measurements are necessary to deduce the dry molar fraction of the gazes of interest (LI-COR) or to calculate the soil evaporation rate (Zawilski, 2022). SCD30 (Sensirion AG, Stäfa, Switzerland) is a cost-effective NDIR sensor that provides $CO_2$ concentration along with air temperature and air

humidity. However, these two last parameters are already provided by BME280 with better accuracy. Both oxygen sensors, LuminOx and OXYBase, were tested, but only one at a time should be used. LuminOx is also measuring the air pressure. However, once again, BME280 is providing air pressure with very good precision that may be used for both sensors.

## 3 Sensors test

Embedded sensors such as SCD30, for $CO_2$, LuminOx, and OXYBase, for $O_2$, concentration determination was truly checked

by cross-measurement with a reference sensor, if available, or using a reference experimental measurement (respiration).

## 3.1 SCD30 cross test

Before using the SCD30 sensor for $CO_2$ monitoring, it was tested by comparison with the high-precision optical feedback-cavity enhanced absorption spectroscopy (OF-CEAS) Li-7810 from Li-Cor (LI-COR Biosciences, Nebraska, USA) for three

months using six chambers. To avoid any difference between measurements due to the air-leading pipes, we installed SCD30 and Li-840A (LI-COR Biosciences, Nebraska, USA) close to the Li-7810 in the same external circuit. Figure 7 shows all the measurements of Li-7810 versus SCD30. A linear regression of these measurements shows a good correspondence with a 1.08 slope and a small offset of less than 27 ppm with a rather high correlation coefficient ($R^2$=0.98). It is worth noting that SCD30 exhibits much better correspondence with Li-7810 than our flow-through LI-840A Infra-Red Gas Analyzer (IRGA), which is

not self-calibrated and, probably, quickly deserves a deep cleaning despite the air filter presence (Fig. 8).





**Figure 7. OF-CEAS Li-7810 measurements versus NDIR SCD30 measurements during a three-month campaign conducted with six chambers. The red solid line represents a linear regression.**





## 3 month measurement using 6 chambers


**Figure 8. OF-CEAS Li-7810 measurements versus IRGA Li-840A measurements during three months. Li-840A derived during the test and presents measured $CO_2$ saturation for 2000 ppm due to the analog output configuration.**

A good air analyzer is always preferable to a small (or even minuscule) analyzer. However, small analyzers can be embedded

under the cloche when big analyzers can only function outside of the chamber, which induces some other problems such as

air-leading tube disturbances, external condensation, and so on. Also, the price difference between a Li-840A and an SCD30

is about 160-fold, and between a Li-7820 and an SCD30 is about 1300-fold.





However, for some GHG gases, such as $N_2O$, a miniaturized, precise-enough analyzer does not exist. A variant of the described chamber, designed for an external muti-gas Fourier-transform Infrared Spectroscopy (FTIR) GT5000 Terra analyzer (Gasmet

Technologies Oy, Vantaa, Finland), was built. In this case, only the BMP280 and the fan were embedded under the cloche, and the data was transferred to a laptop PC via Bluetooth. Two fittings on the top of the chamber were added for air-leading tubes (In and Out). A detailed discussion about the problems with external analyzers will be presented elsewhere.

Arduino programming is relatively simple. All I²C bus-attached devices have available libraries and programming assistance widely available on the web. UART-bus-based sensor communication is relatively easy to establish. The GPS also has

numerous libraries that can be used, as they generally use the NMEA 0183 protocol.

### 3.2 LuminOx and OXYBase cross tests

To test the oxygen sensors, we performed a comparison between LuminOx and OXYBase sensors performing an apparent

respiration quotient (*ARQ*) measurement test with both sensors at the same time. To test *ARQ* measurement, an animal contained in a closed space would be very helpful. However, any experimentation including an animal is strictly regulated by law. These restrictions do not concern volunteer humans and one of us accepts to be briefly closed in a vinery, clean, pressurized tank of 22 hl volume (2.2 $m^3$). The $CO_2$ and $O_2$ evolution was monitored by a battery-powered data logger reading all available sensors at the same time.


As expected, the $CO_2$ evolution with time is nearly linear:



**Figure 9. Measured $CO_2$ evolution in a tank with a breathing human inside. The red solid line represents a linear regression.**

The $O_2$ evolution measured by LuminOx and OXYBase was close with a small offset and relatively matching slope:

## Respiration

$$y = -1.3842 + 1.0676x \quad R^2 = 0.99526$$

**Figure 10. Measured $O_2$ part with LuminOx versus $O_2$ measured with OXYBase during the respiration experience. The red solid line represents a linear regression.**


The *ARQ* calculation, using a linear regression explained in paragraph 5, determined with OXYBase's measurements is 0.97 when the *ARQ* determined with LuminOx's measurements is 0.90. Both sensors allow then a relatively correct *ARQ* calculation.





**Figure 11. CO₂ part measurements versus O₂ part measured by OXYBase (lower abscissa) and O₂ part measured by LulminOx (upper abscissa). The red solid line represents a linear regression of CO₂ versus O₂ measured by OXYBase.**





## 4 Typical results

A typical measurement with a closed chamber technique displays $CO_2$ rising and $O_2$ decaying with time. Historically, the
effluxes or influxes were calculated using a linear regression on the initial data. However, several authors have shown that linear regression can lead to a severely biased calculation (Kutzbach et al., 2007; Silva et al., 2015), but it is beyond the scope of this note to discuss it here. As an illustration, we use the "exponential rise" regression, also called "asymptotic regression". Figure 9 displays a typical carbon dioxide accumulation measured with a chamber positioned on its collar pressed into the soil.

### Typical CO$_2$ Data

| $y = m1 + m2*(1 - exp(-m3*x))$ | | |
|---|---|---|
| | Value | Error |
| m1 | 345.88 | 4.622 |
| m2 | 4487.3 | 40.525 |
| m3 | 0.00081077 | 1.2238e-5 |
| Chisq | 94607 | NA |
| $R^2$ | 0.9993 | |

**Figure 12. Typical CO₂ measures in the chamber. The solid red line represents an asymptotic regression (fit).**

The calculated $CO_2$ efflux $F_{CO2}$ would be then:





$$F_{CO2} = m2 * m3 * R$$

230                                                                                                                              (1)

With *m2* and *m3* being the curve regression constants calculated using a plot of $CO_2$ concentration versus closing time (Fig. 12) and *R* being the actual volume-to-surface ratio. The measured asymptotic concentration is given by the sum *m1+m2* and represents the $CO_2$ concentration in the superior soil layer (0.5% here).

A similar calculation could be conducted on the $O_2$ concentration to determine the oxygen influx.

Figure 13 displays a typical $O_2$ measurement taken at the same time as the $CO_2$ concentration in Fig. 12.





## Typical O$_2$ Data



| y = m1 -m2*(1- exp(-m3*x)) | | |
|---|---|---|
| | Value | Error |
| m1 | 2.0268e+5 | 39.874 |
| m2 | 18917 | 214.21 |
| m3 | 0.0009917 | 2.1601e-5 |
| Chisq | 6.4738e+6 | NA |
| R$^2$ | 0.99786 | |

**Figure 13. Oxygen concentration measurement versus time. The solid red line represents an asymptotic regression.**

Similar to $F_{CO2}$ calculations, the corresponding oxygen influx $F_{O2}$ calculation would be:

$F_{O2} = m2 * m3 * R$

$$(2)$$

With *m2* and *m3* being the constants deduced from an asymptotic regression of the plot of O$_2$ concentration versus time and *R* being the same volume-to-surface ratio as for the $F_{CO2}$ calculation in Formula 1. Always similar to the asymptotic CO$_2$





concentration, the measured asymptotic $O_2$ concentration is given by the difference *m1-m2* and represents the $O_2$ concentration
in the superior soil layer (18.4% here).

## 5 Apparent respiration quotient circulation

To determine the apparent respiration quotient (*ARQ*), by its definition, we can proceed with a quotient of $F_{CO2}$ and $F_{O2}$ formation.

$$ARQ = \frac{F_{CO2}}{F_{O2}}$$

(3)

This quotient comes from the definition of the *ARQ* ($CO_2$ flux divided by $O_2$ influx).

In the reported typical measurements, using this quotient, *ARQ* = 0.194

However, this calculation accumulates uncertainties in both non-linear regressions used for $F_{CO2}$ and $F_{O2}$ determinations.
Another simple way to calculate *ARQ* would be to use $CO_2$ concentration versus $O_2$ concentration, then a linear regression to give direct *ARQ*. Indeed, if we suppose that *ARQ* is constant during the time of the chamber closure, we can write:

$$ARQ = \frac{\frac{dCO_2}{dt}}{-\frac{dO_2}{dt}}$$

(4)

Then:

$$\frac{dCO_2}{dt} = -ARQ * \frac{dO_2}{dt}$$

(5)

By integration

$$CO_2(t) = -ARQ * O_2(t) + C_0$$

265 (6)

*With $C_0$* being a constant:

$$C_0 = ARQ * O_2(t = 0) + CO_2(t = 0)$$

(7)


Figure 14 displays the typical $CO_2$ and $O_2$ measurements already shown in the previous figures, but this time the $CO_2$ concentration is plotted versus the $O_2$ concentration. A linear regression provides simple *ARQ* determination.



## Typical ARQ calculation

Figure 14. CO₂ concentrations versus O₂ concentrations. The solid red line represents a linear regression.

We can note that the *ARQ* provided by the linear regression of $CO_2$ concentrations versus $O_2$ concentrations (*ARQ* = 0.212) is slightly different from the *ARQ* calculated using the $F_{CO2}$ and $F_{O2}$ quotients (8.5% difference). For *ARQ* determination, we suggest using a $CO_2$ concentration versus $O_2$ concentration plot and a linear regression, as it does not accumulate successive non-linear regression uncertainties. We may also note that the measurements done with this chamber using low-cost sensors provide an excellent confidence level $R^2$ matching the theoretical linear and asymptotic regression.

We can also note that the expected *ARQ* would be close to 1, as for respiration, the same amount of $O_2$ is absorbed as the quantity of $CO_2$ released. However, this does not account for the fact that the soil may capture and store a consequent amount of the produced $CO_2$ (Sánchez-Cañete et al., 2018).

## 6 Conclusion

The importance of the soil's most significant natural $CO_2$ production measurement does not have to be proved. For this purpose, an ultra-low-cost portable chamber was built and is described in this note with the hope of helping our scientific community develop their own devices. The described chamber uses only commercial parts with little mechanical work. All used sensors are digital and cost-effective yet accurate enough to allow measurements with excellent confidence level $R^2$ when regressed to adequate linear and non-linear laws. The described chamber is easy to build and easy to operate, allowing a wide range of

users to work with it.

*Disclaimer*

No human was harmed during these studies.

*Authors contributions*

BZ conceived the portative chamber, found and tested the sensors, and wrote the first draft.

VB defined the specifications, found the necessary budget, and reviewed the draft.

*Competing interests*

The contact author has declared that none of the authors has any competing interests.

*Acknowledgments*

We would like to acknowledge Anna Schmid (Precision Sensing GmbH, Germany) for her assistance during the OxyBase sensor installation and its functioning. We are grateful to Gregor Christandl (Elyte Diagnostics GmbH, Austria) for his help

during the Arduino programming.

*Financial Support*

This development was financed by the SEPSOL project, held by the Continental and Coastal Ecosphere Structuring Initiative (EC2CO), a program coordinated by the INSU and supplemented by two other CNRS institutes: the INEE in particular and

the INC.



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
