# Peer review of "Ultra-low-cost manual soil respiration chamber"

_EGUsphere, 2023_

## Author Response (AR1)

**We are grateful to our referees RC1, and RC2 for their constructive comments that help us improve our paper. Here, we answer (in bold), point-by-point, with raised points.**

**Comments and answers to RC1**

This paper presents a low-cost setup to measure gas in the field. The technical details are presented to enable the reader to reproduce the chambers for its own use, enabling to realize extensive field sampling strategies at low-cost. In additions it offers a benchmark of different sensors, highlighting the importance to choose reliable ones and use them in an appropriate manner. This paper would be thus of good benefit for the scientific community.

The reviewer did not have major comments implying major/minor revision, but highlighted small points that could be refined. Please find below specific comments:

Line 9: "." missing after "chamber"

**"." Added.**

Line 12: "The wide interest aroused by this construction among our colleagues pushes us to share our achievements.": I would remove this sentence, the interest for such publication is underlying and does not need further highlight.

**Sentence removed.**

Line 16: Precise in which form CO2 is in the soil (gas, sorbed, dissolved?)

**The sentence was corrected as it was inexact. The soil contains the major carbon part of the Earth, not CO2.**

Lines 18-19: Is their more recent work on this, if so please provide newer references.

**We added some recent references**

Lines 121-122: "NDIR sensors have the best quality-to-cost ratio": please provide numbers for NDIR and two other examples, both for cost and quality. Especially which parameter did you use as indication for quality?

**We added some arguments as to why the non-NDIR sensors we found were discarded. Our final choice is based on our classification, which is indicated in the revised version of the paper. However, we don't intend to impose any classification.**

Line 129: "Both are optical, and their functioning is close to each other": remove, this is redundant with next lines.

**Sentence removed.**

Line 135: "however their use is still optical": I would remove this sentence, I don't really understand its meaning.

**The end of the concerned sentence is "however, their use is still *optional*." Most of the soil respiration chambers are not monitoring the oxygen. We think that oxygen measurement is interesting, but the necessary sensors are expensive, so the described low-cost chamber may be used without (optional) oxygen sensors, keeping its utility.**

Line 136: "used, and some of the existing sensor's specifications are summarized in Table 1.": missing a setence.

**Missing a determiner "The" at the beginning of the sentence.**

**Comments and answers to RC2**

The authors present a rather detailed description of a low-cost gas exchange measurement setup with enough instructions for many researchers and especially technicians to reproduce the setup. It is a nice contribution because low-cost systems are needed in order to enable large-scale point measurements of soil respiration, a major part of the global $CO_2$ cycle.

However, for the measurements to be meaningful, the systems need to be reliable and proved to be so, and the uncertainty inherent in each measurement system needs to be quantified. There is some attempt at this in the manuscript, but uncertainty in the actual product of interest, the flux measurements, is not quantified in any way. There is also possibly improper use of certain citations, which needs to be checked.

Point comments and suggestion for a further experiment follows:

- L16: the soil doesn't contain much $CO_2$ as such, but carbon in the form of organic molecules.

**Indeed, our sentence was wrong; the soil contains a major carbon part on the Earth, not necessarily in $CO_2$ form. However, all carbon contents are potentially oxidable to $CO_2$.**

- L135: the Turcu article concerns subsurface gradient O2 and CO2 concentration measurements, not above soil measurements

**Tarcu et al. developed a soil gradient technique for continuous measurement of gaseous *soil fluxes*. They claim that a standard chamber-based measurement is not sufficient; however, it does not mean that oxygen flux measurement is not interesting, and they claim the contrary. We are using this reference to support our sentence stating that oxygen flux measurement "is bringing some interesting information." Even if we can discuss the sufficiency of the chamber-based measurement, these measurements are not useless.**

; the Helm article concerns gas exchange between tree stems and the atmosphere. I'd reconsider the relevance of these references to the issue at hand.

**Helm et Al. describe an ultra-low-cost, closed chamber-based device with an NDIR sensor for $CO_2$ monitoring and a quenching-based sensor for $O_2$ monitoring. The purpose of this device is to monitor steam respiration. We developed an ultra-low-cost, closed chamber-based device with an NDIR sensor for $CO_2$ monitoring and a quenching-based sensor (OXYBase) for $O_2$ monitoring. The purpose of this device is to monitor *soil* respiration. The objects of the studies are different, and the mechanical setup is also different, but the overall technique and used sensor techniques are the same. We believe that it would be a deontological fault to not mention Helm's paper. For the sake of clarity, we deleted the "soil" word from the concerned sentence to explicitly enlarge the field of interest and not limit our statement to soil respiration.**

- L136: the start of this sentence is missing?

**Indeed, a determiner "The" at the beginning of the sentence was missing. The correct sentence starts with "The used..."**

- Table 1: the accuracy of the CO2 sensors is quite bad!

**The *absolute* accuracy of a miniaturized sensor such as NDIR is worse than the accuracy of IRGA or laser-based devices, which are much bigger and more expensive. This point is rather expected. However, what is important for concentration variation measurement is relative accuracy, not *absolute* accuracy. Also, any statement such as "good" or "bad" requires an implicit or explicit goal. Specifically, concerning the NDIR sensors in a chamber, the goal is to measure the relative carbon dioxide increase in a**

**closed chamber over the soil surface. Typically, this evolution is from 400 ppm to over 1000 ppm after several minutes. That is an increase of over 600 ppm. The typical NDIR absolute accuracy is 30 ppm, which is then less than 5% of the relative increase, or less than 3% of the absolute final value. This accuracy is worse than a usual IRGA, which is bigger and requires the use of air-leading tubes along with a pneumatic multiplexer. Air-leading tubes induce some errors due to the air mixing inside the tubes, imprecise timing of the chamber closure delayed by the time of flow through the air-leading tubes, water condensation inside the tubes, and possible $CO_2$ absorption-desorption. Multiplexers are a source of leaks, breakdowns, and so on. Also, the biggest analyzer means the biggest air-analyzing cell volume and corresponding errors. Some errors were already signaled in our previous paper: https://gi.copernicus.org/articles/11/163/2022/gi-11-163-2022.pdf and others will be reported in our next paper. We can only mention here that just an imprecise timing of chamber closure may induce over 10% flux calculation error. As always, each variant has its pros and cons, and it seems difficult to condemn NDIR sensors just because of their worst absolute accuracy.**

- Section 3.1: the results of the cross-testing are unfortunately not very convincing. A correlation between the CO2 readings is not enough to prove the worth of the cheaper sensor; we should see how the measured respiration values fit against each other. From the accompanying figure (Fig. 7) we can see that there are times when the reading of the SCD30 has changed a lot while the reading of the Li-7810 has remained stable. This tells us that the SCD30 has seen a flux while the Li-7810 has not. Please present further analysis of the data by comparing the calculated fluxes between the devices.

**Indeed, in Figure 7, we can see a few cycles when the SCD30 measurements are increasing and when the Li-7810 measurements are frozen. This point comes from the Li-7810 communication issue, as this analyzer sends a very large tram on the only available Ethernet port, saturating the buffer. This issue was solved later during the measurement campaign by increasing drastically the buffer size, but the corresponding data was left as the data was not filtered. In the revised version of our paper, we removed the frozen Li-7810 data and presented one chamber flux calculation using SCD30 data and Li-7810 data for the concerned period.**

- in order to properly evaluate the usefulness of the system, an estimation of the minimum measurable flux is required: set the system to measure a (known) constant gas mixture for a prolonged time, calculate and report the SD of the CO2 readings, and calculate fluxes using random starting points and variable measurement lengths ("closure lengths"), and report using statistical methods how much the observed fluxes vary when the actual change in CO2 concentration is zero. This will give an estimate of the uncertainty in the system and indicate how small fluxes can reliably be observed with it.

We are grateful for this suggestion, and we are including the corresponding measurement in the revised version of our paper. We have limited the "closure length" to the interval from one minute to twenty minutes, as it is the real interval of the chamber operation. As the wanted variable is the flux, we performed some statistical analyses on the calculated carbon dioxide flux. We keep in mind that the flux measurements are not very basic and require some diligence. In the case of small effluxes, a longer closure time would be adopted, and the most significant closure time criterion is the overall measured gas concentration variation amplitude. Of course, the closure time cannot be too long to not excessively perturb measurement conditions.